# Biased versus Partial Agonism in the Search for Safer Opioid Analgesics

**DOI:** 10.3390/molecules25173870

**Published:** 2020-08-25

**Authors:** Joaquim Azevedo Neto, Anna Costanzini, Roberto De Giorgio, David G. Lambert, Chiara Ruzza, Girolamo Calò

**Affiliations:** 1Department of Biomedical and Specialty Surgical Sciences, Section of Pharmacology, University of Ferrara, 44121 Ferrara, Italy; zvdjmg@unife.it (J.A.N.); g.calo@unife.it (G.C.); 2Department of Morphology, Surgery, Experimental Medicine, University of Ferrara, 44121 Ferrara, Italy; anna.costanzini@unife.it (A.C.); dgrrrt@unife.it (R.D.G.); 3Department of Cardiovascular Sciences, Anesthesia, Critical Care and Pain Management, University of Leicester, Leicester LE1 7RH, UK; dgl3@leicester.ac.uk; 4Technopole of Ferrara, LTTA Laboratory for Advanced Therapies, 44122 Ferrara, Italy

**Keywords:** opioids, mu receptor, analgesia, opioid side effects, biased agonism, partial agonism

## Abstract

Opioids such as morphine—acting at the mu opioid receptor—are the mainstay for treatment of moderate to severe pain and have good efficacy in these indications. However, these drugs produce a plethora of unwanted adverse effects including respiratory depression, constipation, immune suppression and with prolonged treatment, tolerance, dependence and abuse liability. Studies in β-arrestin 2 gene knockout (βarr2(−/−)) animals indicate that morphine analgesia is potentiated while side effects are reduced, suggesting that drugs biased away from arrestin may manifest with a reduced-side-effect profile. However, there is controversy in this area with improvement of morphine-induced constipation and reduced respiratory effects in βarr2(−/−) mice. Moreover, studies performed with mice genetically engineered with G-protein-biased mu receptors suggested increased sensitivity of these animals to both analgesic actions and side effects of opioid drugs. Several new molecules have been identified as mu receptor G-protein-biased agonists, including oliceridine (TRV130), PZM21 and SR–17018. These compounds have provided preclinical data with apparent support for bias toward G proteins and the genetic premise of effective and safer analgesics. There are clinical data for oliceridine that have been very recently approved for short term intravenous use in hospitals and other controlled settings. While these data are compelling and provide a potential new pathway-based target for drug discovery, a simpler explanation for the behavior of these biased agonists revolves around differences in intrinsic activity. A highly detailed study comparing oliceridine, PZM21 and SR–17018 (among others) in a range of assays showed that these molecules behave as partial agonists. Moreover, there was a correlation between their therapeutic indices and their efficacies, but not their bias factors. If there is amplification of G-protein, but not arrestin pathways, then agonists with reduced efficacy would show high levels of activity at G-protein and low or absent activity at arrestin; offering analgesia with reduced side effects or ‘apparent bias’. Overall, the current data suggests—and we support—caution in ascribing biased agonism to reduced-side-effect profiles for mu-agonist analgesics.

## 1. Introduction

Opioid analgesics remain the gold standard for the treatment of moderate to severe pain. This is due to their unique mechanism of action; a powerful inhibitory effect both on nociception and on the emotional, cognitive and behavioral responses to pain states. However, the use of opioid analgesics is limited by their significant side effects, which include respiratory depression, constipation and, with prolonged treatment, tolerance, dependence and abuse liability. The right balance between control of pain and the risks associated with opioid drug treatment (particularly with long term treatments) is not easy to achieve. There are countries in which health systems overestimate the risks associated with opioid drug therapies often causing unsatisfactory management of pain (e.g., Italy [1]), whereas the health systems of other countries (e.g., USA) underestimated the risks associated with opioid drug prescription contributing to the opioid epidemic [2] that caused a 4-fold increase of fatal overdoses in the last two decades. This underscores the need for novel drugs that maintain the analgesic effectiveness of classical opioids, but with improved side effect profile.

Different strategies have been developed in the search for safer opioid analgesics, including increasing endogenous opioid signaling with enkephalinase inhibitors [3,4] use of mu opioid receptor positive allosteric modulators [5,6] peripherally restricted opioids [7] or pH-dependent mu-receptor agonists [8,9] and mixed opioid receptor agonists [10,11]. In addition, mixed agonists for mu and nociceptin/orphanin FQ receptors, showing promising profiles in preclinical studies, have been recently reported in the literature (reviewed in [12]). The most advanced among these compounds, cebranopadol [13] is now in advanced clinical development as an analgesic [14,15].

Another potential strategy for the development of safer opioid analgesics is based on the concept of functional selectivity or biased agonism, which is the ability of some receptor ligands to selectively stimulate one signaling pathway (e.g., selectivity for G protein or arrestin) [16]. This phenomenon has great potential in terms of drug discovery since it can be exploited to dissect the various responses associated with the activation of a given receptor. This would facilitate the discovery of ligands able to activate the signaling pathways associated with beneficial effects while avoiding those pathways associated with side effects, thus generating safer drugs [16]. In recent years, biased ligands were identified and characterized for several different G protein coupled receptors (GPCR) [17] including the mu opioid receptor. The aim of this short review is the critical analysis of the available literature regarding the potential of mu-receptor-biased agonists for development as innovative analgesics.

## 2. Genetic Studies

The first indication of a role of β-arrestin 2 in the in vivo regulation of the analgesic response to opioids was provided by Bohn and collaborators with the use of mice knockout for the βarrestin 2 gene (βarr2(−/−)) [18]. In these mice the analgesic effects of morphine are not only preserved, but potentiated; in fact, the ED_50_ of morphine is 10 and 6 mg/kg in βarr2(+/+) and βarr2(−/−) mice, respectively. Moreover, the effects of a single dose of morphine were prolonged in βarr2(−/−) mice. Naloxone prevented the analgesic response to morphine in both genotypes. Finally, no changes in [^3^H]naloxone binding in various brain regions were evident between βarr2(+/+) and βarr2(−/−) mice [18]. Another study [19] demonstrated that βarr2(−/−) mice do not develop tolerance to the analgesic effect of morphine while they were similar to wild type animals in terms of morphine physical dependence, as demonstrated behaviorally by naloxone precipitated withdrawal syndrome and biochemically by upregulation of adenylyl cyclase activity [19]. In addition, the rewarding properties of morphine (but not of cocaine), assessed using the conditioned place preference test, were larger in βarr2(−/−) than βarr2(+/+) mice [20]. The logical next step of the investigation of morphine responses in βarr2(−/−) mice was to study the most common acute side effects of opioid drugs; constipation [21] and respiratory depression [22,23]. As far as opioid-induced constipation is concerned the effects of morphine were investigated in βarr2(−/−) and βarr2(+/+) mice by measuring accumulated fecal boli and bead expulsion time. In both the assays βarr2(−/−) animals were less sensitive to morphine than βarr2(+/+) mice [24]. Similar results were obtained investigating morphine-induced respiratory depression in whole-body plethysmography studies; the results of these experiments demonstrated that morphine produces significantly less suppression of respiratory frequency in βarr2(−/−) mice [24]. These findings were interpreted assuming that βarr2 acts (as expected) as a desensitizing element of morphine analgesia, while it significantly contributes to the cellular signaling relevant for the respiratory and gastrointestinal side effects of morphine [24]. Of note is that these findings contrast with more recent observations demonstrating that opioid-induced respiratory depression is due to mu receptor/G_i_/G protein-coupled inwardly rectifying potassium (GIRK) channels signaling in neurons of the respiratory center [25,26].

Collectively the studies by the Bohn group suggested that eliminating βarr2 increased morphine analgesic potency while decreasing its ability to induce constipation and respiratory depression; this makes morphine a safer analgesic. These studies led to the very attractive hypothesis that drugs able to promote mu receptor interaction with G protein, but not βarr2 (i.e., mu receptor G protein-biased agonists, see next section), should mimic the profile of morphine in βarr2(−/−) mice and could be developed as an innovative class of safer opioid analgesics [27,28].

However recent research findings obtained with genetic tools questioned the above hypothesis. In fact a consortium of three different laboratories in Sydney, Bristol and Jena reexamined opioid side effects in βarr2(−/−) mice [29]. In these studies, three independent groups investigated the respiratory depressant effects of morphine using different plethysmography systems in independently bred βarr2(−/−) and βarr2(+/+) mice. In all three sets of results morphine, in the range of doses 3–30 mg/kg, produced a dose dependent reduction of respiratory rate which was virtually superimposable in βarr2(−/−) and βarr2(+/+) animals. Similar results were obtained by the group in Jena using fentanyl (0.05 –3 mg/kg) as the mu-receptor agonist. In addition, the same group also reinvestigated opioid-induced constipation in βarr2(−/−) and βarr2(+/+) mice. Both morphine and fentanyl elicited a dose dependent reduction of accumulated fecal boli with similar potency and maximal effects in βarr2(−/−) and wild type animals [29]. We have performed similar experiments and our findings are summarized in Figure 1 and Figure 2. In line with Bohn’s original findings [18], morphine (0.1–10 mg/kg) and fentanyl (0.01–1 mg/kg) elicited dose dependent antinociceptive effects in the mouse tail withdrawal assay being approximately two-fold more potent in βarr2(−/−) than βarr2(+/+) mice (Figure 1). In accumulated fecal boli experiments, morphine (3–30 mg/kg) and fentanyl (0.1–1 mg/kg) dose dependently inhibited gastrointestinal functions with no major differences in βarr2(−/−) compared to βarr2(+/+) mice (Figure 2), which is in agreement with the findings obtained by the group in Jena [29]. The reason for the discrepancy between the results obtained with βarr2(−/−) mice by different research groups is not known; it has been suggested [29] that mixed genetic backgrounds may have a role in the discrepancy. However, in all studies, this possible confounding factor was considered and minimized using littermates or backcrossed animals.

In a recent very elegant study novel genetic tools have been generated and investigated in order to shed light on the relationship between arrestins and morphine analgesia and side effects [30]. As described for virtually all GPCR [31], the activated mu receptor is recognized by G protein–receptor kinases (GRKs) that phosphorylate several serine and threonine residues located in the cytoplasmic loops and carboxyl-terminal; the phosphorylated receptor can then bind arrestins. To prevent this phenomenon three lines of mutant mice were generated by knocking in mu receptor genes with serine- and threonine-to-alanine mutations in the carboxyl-terminus of the protein that render the receptor increasingly unable to recruit β-arrestins. Thus, these mutant mice express G-protein-biased mu receptors. Importantly, autoradiographic studies demonstrated no differences between the knock-in lines and wild type animals in terms of mu receptor density in different brain areas. The phosphorylation-deficient mu knock-in mice displayed; (i) enhanced opioid-mediated analgesia in the hot-plate test, (ii) reduced liability to develop tolerance to the analgesic effects of opioid drugs after a seven-day chronic treatment with osmotic pumps and (iii) similar signs of withdrawal in response to the administration of naloxone after chronic treatment with morphine or fentanyl, all compared to wild type animals. With respect to these opioid related actions, these knock-in mice displayed a phenotype similar to that of βarr2(−/−) mice [18,19]. These data demonstrated that mu receptor carboxyl-terminal multisite phosphorylation and mu receptor/βarr2 interaction are crucial regulators of opioid analgesia and tolerance, but not physical dependence. As far as the respiratory and gastrointestinal side effects of opioids are concerned, all genotypes of phosphorylation-deficient G protein-biased mu knock-in mice responded to equianalgesic doses of morphine and fentanyl with profound respiratory depression and constipation. Moreover, a detailed analysis of morphine and fentanyl ED_50_ values for analgesia versus their ED_50_ for respiratory depression and constipation yielded highly significant correlation coefficients. Collectively these findings suggest that the lack of mu receptor phosphorylation promotes enhanced analgesia and a proportional increase in respiratory depression and constipation thus not supporting a role for β-arrestin signaling in opioid side effects [30]. Clearly these results argue against the hypothesis that mu agonists biased toward G proteins may act as safer analgesics.

## 3. Pharmacological Studies—Are Mu-Receptor Agonists Biased Toward G Proteins Safer Analgesics?

Based on the original findings obtained in βarr2(−/−) animals [18,19,24] and on the hypothesis that mu receptor/βarr2 signaling is involved in opioid acute side effects, several groups developed projects aimed at the identification and pharmacological characterization of mu receptor G protein-biased agonists as innovative analgesics.

The first and most widely studied molecule of this class is oliceridine (aka TRV130) [33] (see chemical structure in Figure 3). The structure–activity relationship study that led to its identification is described by Chen and colleagues [34]. Oliceridine binds to the human, rat and mouse mu receptor with low nanomolar affinities and inhibits cAMP accumulation in HEK293 cells expressing the human recombinant mu receptor with similar maximal effects but higher potency than morphine. However, oliceridine displayed a lower efficacy than morphine for stimulating mu receptor phosphorylation and internalization and for recruiting βarr2. Moreover, the inhibitory effects of oliceridine in the cAMP assay were competitively antagonized by naloxone while oliceridine competitively antagonized DAMGO-induced βarr2 recruitment. Of note the agonist potency of oliceridine in the cAMP assay (pEC_50_ 8.2) is similar to its antagonist potency (pA_2_ 7.7) in βarr2 recruitment experiments. Thus, in vitro studies suggested that oliceridine behaves as a mu-receptor agonist biased toward G proteins [35]. In the same study DeWire et al. investigated the in vivo actions of oliceridine. This compound elicited dose dependent and robust antinociceptive effects in various analgesiometric assays in mice and rats producing similar maximal effects to morphine but with approximately 10-fold higher potency. Interestingly, in experiments investigating the gastrointestinal (fecal boli accumulation and glass bead expulsion in mice) and respiratory (blood pCO_2_ and pO_2_ in rats) side effects of opioids, oliceridine was less potent and effective than morphine [35]. Thus, compared to morphine, the G protein-biased mu receptor agonist oliceridine displayed a larger therapeutic index; this finding corroborates the hypothesis based on the original studies in βarr2(−/−) animals [18,24] that βarr2 signaling is involved in the acute side effects of opioid analgesics. The G protein-biased mu receptor agonist activity of oliceridine as well as its antinociceptive activity were later confirmed by Mori et al. [36]. In this study, the authors also demonstrated that the antinociceptive effects of oliceridine in the mouse sciatic nerve ligation model are associated with lower tolerance liability than fentanyl [36]. This latter finding was independently confirmed by a different group that compared the antinociceptive effects of morphine and oliceridine after a 4-day treatment with the mouse tail withdrawal assay [37]. Interestingly these authors also reported that mice treated chronically with oliceridine display, in response to an injection of naloxone, a withdrawal syndrome similar to that observed in morphine treated mice. Again, these findings are in line with the original hypothesis of the Bohn group [19] that βarr2 signaling is involved in opioid tolerance, but not dependence.

As far as abuse liability is concerned, oliceridine has been investigated in rats self-administering drugs under a progressive-ratio schedule of reinforcement. Compared to oxycodone, oliceridine was found to be equipotent and equieffective in self-administration and thermal antinociception experiments [38]. Similar results were reported in rat fentanyl discrimination studies where oliceridine was approximately two-fold more potent in producing fentanyl stimulus effects versus antinociception [39]. The abuse liability of oliceridine was also reported by Altarifi et al. [40] using an intracranial self-stimulation procedure in rats. This study also questioned the increased therapeutic index of oliceridine when compared to morphine. Indeed, the effects of oliceridine were the same as morphine both in the mouse tail withdrawal assay and in the mouse accumulated fecal boli test [40].

Oliceridine was also used in comparison with morphine [41,42], or other mu ligands [43,44] in molecular dynamics studies aimed at investigating the active structure of mu receptor interacting with G protein and arrestin. However, the detailed analysis of these studies goes beyond the scope of this article.

Oliceridine has been investigated in the clinic. A first-in-human study was conducted with ascending doses of oliceridine administered intravenously over the dose range of 0.15–7 mg [45]. Oliceridine caused dose-related pupil constriction confirming mu receptor engagement. Nausea and vomiting observed at the 7 mg dose limited further dose escalation. Collectively this study suggests that oliceridine may have a broad margin between doses causing mu receptor-mediated pharmacology and doses causing mu opioid receptor-mediated intolerance [45]. Another study investigated the effects of oliceridine and morphine after single intravenous injections in thirty healthy men; oliceridine produced greater analgesia than morphine with less reduction in respiratory drive and less severe nausea [46]. The efficacy and tolerability of oliceridine in acute pain management after bunionectomy has been investigated in a Phase II, randomized and placebo- and active-controlled study. The results demonstrated that oliceridine rapidly produces profound analgesia in moderate to severe acute pain, with a profile of tolerability like morphine [47]. Similar results were obtained in a Phase IIb study in patients with moderate to severe acute pain following abdominoplasty. These clinical results suggest that oliceridine promotes effective, rapid analgesia in patients with postoperative pain, with acceptable safety/tolerability profiles and a potentially wider therapeutic window than morphine [48]. Moreover, favorable analgesia over respiratory depression has been reported for oliceridine, but not morphine in a recent study that reanalyzed data obtained from healthy volunteers and postoperative patients [49] Finally, the analgesic effectiveness and favorable safety/tolerability profiles of oliceridine regarding respiratory and gastrointestinal adverse effects compared to morphine have been confirmed in different Phase III studies [50,51,52]. In October 2018, the FDA Advisory Committee voted 8 against and 7 in favor of the approval of oliceridine for the management of moderate to severe acute pain. A new application was submitted and on the seventh August 2020 the FDA approved oliceridine with the name Olinvyk™ for short term intravenous use in “hospitals and other controlled settings”. Oliceridine will soon be available in the market and larger studies will more thoroughly define its analgesic effectiveness and tolerability profiles in clinical practice.

Another interesting molecule acting as G-protein-biased mu-receptor agonist is PZM21 [53] (see chemical structure in Figure 3). This molecule was identified by docking over three million commercially available lead-like compounds with the orthosteric pocket of the 3D crystal structure of the mu receptor; solved in its inactive state in 2012 [54]. In receptor binding studies PZM21 displayed high affinity (pK_i_ 9) and selectivity for the mu opioid receptor. In functional studies PZM21 behaved as a potent agonist in different Gi/o-mediated signaling assays while it was virtually inactive in recruiting βarr2. Agonist activity of PZM21 in the βarr2 recruitment assay can be detected after transfecting cells with GRK2. However, under these conditions, the efficacy of PZM21 (0.32) was a fraction of that of DAMGO (1.00) and even morphine (0.52) [53]. Thus, these results suggest that PZM21 behaves as a mu receptor agonist biased toward G protein signaling. Of note in all these experiments oliceridine has been also assayed, producing results similar to those obtained with PZM21. When tested in mice, PZM21 (10–40 mg/kg) elicited antinociceptive effects in the hotplate and formalin test, but not in the mouse tail flick assay. The reason for these assay-specific effects of PZM21 are presently unknown. More important, the analgesic effects of PZM21 (as well as those of morphine) in the hotplate test were absent in mu receptor gene knockout mice. When tested in the mouse accumulated fecal boli assay at equianalgesic doses PZM21 elicited constipation that was less than that produced by morphine. Concerning respiratory effects, whole-body plethysmography studies demonstrated that while morphine profoundly depressed respiration, the effects of PZM21 was indistinguishable from vehicle [53]; however it should be noted that in these experiments the vehicle injection produced a respiratory depressant effect. Collectively, the results obtained with PZM21 confirm findings with oliceridine suggesting that mu receptor agonists biased toward G protein signaling elicit robust analgesia associated with less respiratory and gastrointestinal side effects.

However, some PZM21 findings were not confirmed in other studies. In bioluminescence resonance energy transfer (BRET) studies PZM21 behaved as a low efficacy partial agonist in promoting mu receptor interaction with both G protein and βarr2 [55]. The analgesic effect of PZM21 in the hot plate assay has been confirmed but this was associated with a clear inhibition of respiratory function. In mice receiving twice-daily doses of PZM21 for four days, complete tolerance developed to the antinociceptive but not respiratory depressant effects [55]. Similar findings were previously obtained with morphine in a study performed under the same experimental conditions [56]. Thus, in this latter study, no major differences were found between the pharmacological profile of PZM21 and that of the standard opioid analgesic morphine. PZM21 caused dose-dependent antinociception after systemic and spinal administration; after repeated administration tolerance developed to the antinociceptive actions of PZM21 and animals became physically dependent as demonstrated by naloxone-precipitated withdrawal syndrome [57]. Recently the actions of PZM21 were compared with those of morphine and oxycodone in non-human primates [58]. After systemic administration, PZM21-induced dose-dependent thermal antinociceptive effects being 10-fold less potent than oxycodone. In self-administration studies PZM21 exerted reinforcing effects similar to oxycodone. After intrathecal administration, PZM21 mimicked morphine, producing naltrexone sensitive antiallodynic effects associated with long-lasting scratching [58]. The chemical template of PZM21 has been used for structure activity relationship studies that led to the identification of novel G-protein-biased mu-receptor agonists [59,60].

Other compounds acting as selective mu-receptor agonists with different degrees of bias toward G protein signaling have been discovered in a study specifically aimed at investigating whether the magnitude of the bias factor of a mu agonist will impact its therapeutic index (analgesia versus respiratory depression) [61]. A series of compounds with a piperidine core structure were generated and demonstrated to act as high affinity and selective mu ligands in receptor binding studies. These novel compounds were investigated using the human recombinant mu receptor in functional assays including agonist stimulated GTPγ[^35^S] binding, cAMP accumulation and βarr2 recruitment using the operational model [62] to estimate their bias factors. The pharmacological effects of these compounds were systematically compared to those of DAMGO as a reference agonist and of morphine and fentanyl as clinically relevant drugs. Results of these studies suggest that fentanyl and SR-11501 behave as mu receptor βarr2-biased agonists, morphine, SR-14968 and SR-14969 behave as unbiased agonists, SR-15098, SR-15099 and SR-17018 (see chemical structure in Figure 3) behave as G protein-biased agonists [61]. Importantly this rank order of bias remains the same when compounds were tested in preparations expressing the mouse mu opioid receptor. Pharmacokinetic studies demonstrated that these SR compounds are able to cross the blood–brain barrier after systemic administration. In the hot plate and tail withdrawal assays SR compounds elicited antinociceptive effects in wild type mice but not in mu receptor gene knockout mice, confirming their high selectivity of action in vivo. When tested for respiratory depressant effects (oxygen saturation and respiratory rate) at morphine equianalgesic doses SR-15098, SR-15099 and SR–17018 (i.e., those molecules showing the higher bias toward G proteins), produced the least respiratory suppression. To carefully estimate the therapeutic index of these molecules dose response studies were performed and ED_50_ values calculated for the analgesiometric assays (hot plate and tail withdrawal tests) and for respiratory depressant effects (oxygen saturation and respiratory rate). A robust correlation between bias factor and therapeutic index was found: the higher the bias toward G protein signaling, the higher the therapeutic index. In a recent study the effects of chronic treatment with SR–17018 via subcutaneous osmotic minipumps was investigated [63]. In contrast to morphine, SR–17018 does not produce tolerance in the hot plate test. However, after minipump removal, mice treated with SR–17018 displayed significant signs of withdrawal, similar to morphine [63].

Collectively the results of these studies appear to confirm the original hypothesis, based on work with βarr2(−/−) mice [18,19,24], that mu-receptor agonists that do not recruit βarr2 display reduced tolerance liability and, more important, are safer analgesics.

## 4. Pharmacological Studies—Are Mu Receptors Partial Agonists Safer Analgesics?

Direct comparison of the pharmacological features of novel molecules investigated in different laboratories, with different in vitro assays and protocols and diverse in vivo models is always difficult. Gillis et al. [64] reexamined the pharmacological profiles of the G protein-biased agonists oliceridine, PZM21 and SR–17018 in parallel experiments and compared the profiles with those of DAMGO and the clinically viable drugs fentanyl, methadone, morphine, oxycodone and buprenorphine. The mu agonist properties of this panel of ligands were carefully examined using rigorous pharmacological approaches, which consisted of using the same cell line (HEK293 cells expressing the human recombinant mu receptor) and a large panel of assays. To investigate mu/G protein pathways BRET-based assays were used to measure mu receptor interaction with a conformationally selective nanobody, with a truncated, soluble “mini” G_i_ protein and with Gα_i2_. Moreover, mu receptor inhibition of cAMP levels via G_i_ was also studied with a BRET assay. In addition, Gβγ-mediated activation of GIRK channels was investigated with a membrane potential-sensitive dye. To investigate mu receptor regulatory pathways BRET-based assays were used to measure mu receptor interaction with GRK2 and βarr2 and also mu receptor internalization. Finally using phosphosite-specific antibodies, agonist-induced C-terminal phosphorylation of the mu receptor was studied. Importantly, in order to obtain robust and consistent concentration–response curves that allow a precise assessment of ligand potency and efficacy, manipulations were performed with the aim of avoiding conditions characterized by an extremely low or extremely high efficiency of the stimulus–response coupling. Thus, GIRK experiments were performed in the absence and presence of an irreversible mu antagonist, βarr2 recruitment and receptor internalization studies were performed in the absence and presence of overexpressed GRK2 and nanobody and mini G_i_ protein recruitment experiments were performed with an excess of reporter probes.

The results obtained in G protein assay demonstrated that oliceridine, PZM21 and SR-17018 are indeed mu receptor partial agonists. In particular the following rank order of maximal effects was determined: DAMGO = fentanyl = methadone > morphine = oxycodone > oliceridine = PZM21 ≥ SR-17018 ≥ buprenorphine which was highly conserved in all the assays (r^2^ always ≥ 0.79) including ligand-induced C-terminal phosphorylation of the mu receptor. Of note is that the partial agonist behavior of PZM21 and oliceridine at the mu receptor has already been reported for ion channel signaling using electrophysiological and Ca^2+^ imaging techniques [65] and in biochemical assays (Azzam et al. personal communication). Surprisingly, this same rank order of maximal effects was measured in receptor regulatory pathway assays (GRK2 and βarr2 and receptor internalization) for all compounds; thus, suggesting that they have similar activity in both G protein and receptor regulatory pathways. These results were confirmed by operational model analysis that demonstrated across the different assays no significant bias factors for all ligands, including, the putative G-protein-biased-agonists oliceridine, PZM21 and SR-17018. For a comprehensive discussion of the possible reasons that may explain the discrepant results obtained by Gillis et al. [64] compared to previously published findings [35,53,61], the reader is referred to Gillis et al. [66].

In order to compare their analgesic and respiratory depressant effects, fentanyl, morphine, oliceridine, PZM21, SR-17018 and buprenorphine were evaluated in dose–response studies in the hot plate and whole-body plethysmography assays [64]. All compounds evoked robust and dose-dependent antinociceptive effects in the hot plate test with kinetics of action in line with previously published findings. However, the dose–response curve for SR–17018 could not be completed due to solubility issues. In whole-body plethysmography dose–response studies, all compound produced a statistically significant reduction in respiratory frequency, but the effects of buprenorphine, oliceridine, PZM21 and SR–17018 were lower than those of morphine or fentanyl. The results of in vivo experiments were then used to calculate the therapeutic index of these mu agonists (of note the therapeutic index of SR–17018 could only be roughly estimated due to incomplete dose–response curve data). The rank order of therapeutic indices was buprenorphine > SR–17018 = PZM21 ≥ oliceridine ≥ morphine ≥ fentanyl. There was no correlation between therapeutic index and bias factor while there was a clear inverse relationship between therapeutic index and ligand efficacy [64]. Importantly the above mentioned rank order of therapeutic index is in line with clinical studies suggesting the following rank order of tolerability for the treatment of moderate to severe pain buprenorphine > morphine > fentanyl [67,68].

Collectively this study demonstrated that putative G-protein-biased agonists behave as low efficacy partial agonists. Moreover, this study confirmed the higher therapeutic index (analgesia versus respiratory depression) of oliceridine, PZM21 and SR–17018 (as well as buprenorphine) compared to classical opioid analgesics (morphine and fentanyl) and provides robust evidence that these actions are likely due to and can be predicted by, partial rather than biased agonism.

## 5. Conclusions

Early studies performed with βarr2(−/−) mice suggested that mu receptor interaction with βarr2 is involved in morphine gastrointestinal and respiratory side effects [24] but not its analgesic action [18]. This observation led to the hypothesis that mu-receptor agonists biased toward G protein may offer safety advantages as analgesics. This hypothesis was later confirmed in preclinical studies demonstrating that the mu receptor G protein-biased agonists oliceridine [35], PZM21 [53] and SR-17018 [61] displayed an improved therapeutic index compared to morphine. For oliceridine this improved therapeutic index has been confirmed in a large series of clinical studies [33]. However this general supposition of improved side effect profile has recently been questioned by the following data: different laboratories did not replicate the original findings regarding the improved therapeutic index of morphine in βarr2(−/−) mice [29], the therapeutic index of morphine is not improved in genetically engineered mice expressing G protein-biased mu receptors [30]. In addition, a recent study in which oliceridine, PZM21 and SR-17018 were tested in parallel in vitro and in vivo experiments confirmed the improved therapeutic indices of these mu ligands but demonstrated that their improved safety profile is likely attributable to low efficacy partial agonism rather than G protein-bias [64]. Based on the available evidence it is reasonable to suggest that the biased agonism as a strategy is unlikely to produce safer opioid analgesics. We close with the following statement from T Kenakin [16]: “biased signaling still has the potential to justify revisiting of receptor targets previously thought to be intractable and also furnishes the means to pursue targets previously thought to be forbidden due to deleterious physiology”. Increasing the rate of success in drug discovery programs based on biased agonism requires rigorous pharmacological approaches to both assay development and data analysis. Moreover, knowledge of cell types responsible for specific pathologies and the associated signaling pathways activated during that pathological insult also require careful study, as discussed by Michel et al. [69].

## Figures and Tables

**Figure 1 molecules-25-03870-f001:**
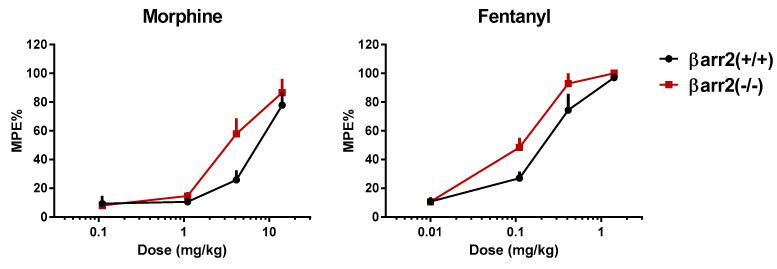
Mouse tail withdrawal assay. Dose response curves to morphine (**left** panel) and fentanyl (**right** panel) in βarr2(+/+) and βarr2(−/−) mice. Data are mean ± SEM of 6 animals for each treatment. Experiments were performed as described in [32].

**Figure 2 molecules-25-03870-f002:**
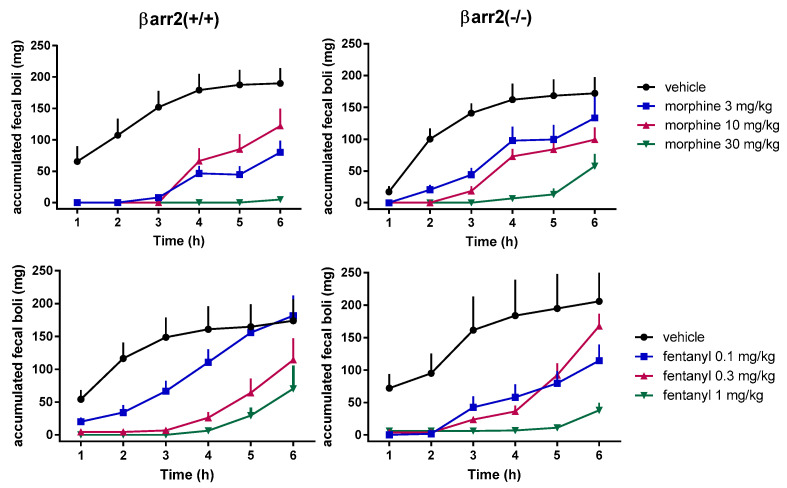
Mouse accumulated fecal boli assay. Dose response curves to morphine (**top** panels) and fentanyl (**bottom** panels) in βarr2(+/+) (**left** panels) and βarr2(−/−) (**right** panels) mice. Data are mean ± SEM of 7 animals for each treatment. Experiments were performed as described in [24].

**Figure 3 molecules-25-03870-f003:**
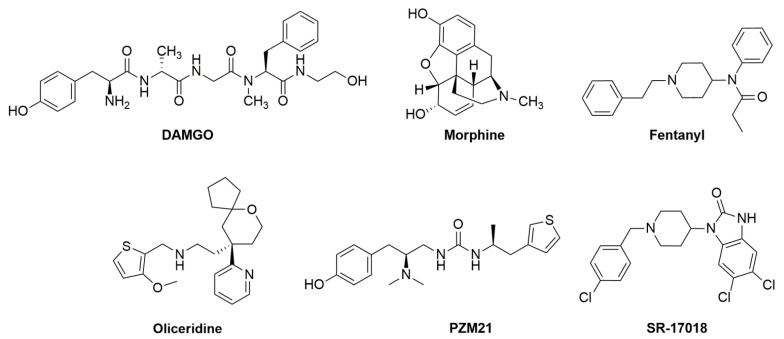
Chemical structures of DAMGO, morphine, fentanyl, oliceridine, PZM21 and SR–17018.

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
