# Peer review of "Biased versus Partial Agonism in the Search for Safer Opioid Analgesics"

_molecules, 2020, doi:10.3390/molecules25173870_

Round 1

Reviewer 1 Report

The Authors did a very good job putting together this Review where they describe the scientific journey that led to the general view of biased agonism of opioid receptors as the next frontier in drug development for analgesic applications.

They start by the discoveries made using the Beta Arrestin2 knock out mice and continue describing the discovery and characterization of 3 classical pharmacological agents with G-protein biased capability at the Mu opioid receptor, namely oliceridine, PZM21 and SR17018. 

Importantly then the Authors focus on the comparison between therapeutic index and drug efficacy or bias factor and point the attention to the fact that only the first two are correlated.

The only minor suggestion to the authors I have is that the last two phrases of the abstract are a bit difficult to read. I suggest to rephrase them for improved flow. 

Author Response

The Authors did a very good job putting together this Review where they describe the scientific journey that led to the general view of biased agonism of opioid receptors as the next frontier in drug development for analgesic applications.

They start by the discoveries made using the Beta Arrestin2 knock out mice and continue describing the discovery and characterization of 3 classical pharmacological agents with G-protein biased capability at the Mu opioid receptor, namely oliceridine, PZM21 and SR17018.

Importantly then the Authors focus on the comparison between therapeutic index and drug efficacy or bias factor and point the attention to the fact that only the first two are correlated.

We would like to thank Reviewer #1 for his/her nice comments to our manuscript.

The only minor suggestion to the authors I have is that the last two phrases of the abstract are a bit difficult to read. I suggest to rephrase them for improved flow.

As suggested by the Reviewer #1 last two phrases of the abstract have been rephrased in the revised version of the manuscript hoping that the flow has been improved.

Reviewer 2 Report

Development of MOR agonists that are efficacious but have less adverse effects are challenging. This manuscript provides a timely summary of the current research data, especially on a few notable new molecular entities, such as oliceridine, PZM21 and SR-17018. The text is clearly written, and literature coverage up to date. The main conclusion and cautious recommendation are well balance. Overall, the manuscript has good educational value.

Since oliceridine is one of the key compounds under discussion, I would urge the authors to wait for the PDUFA announcement from FDA which is due on August 7th and make appropriate revision based on the conclusion and disclosed new data.

Reference 69 is a comprehensive introduction on oliceridine, hence its citation should be moved to earlier place where oliceridine is mentioned rather than the current place in Conclusions.

There are a few references (e.g. 2, 8, 12, 30, 31, 32, 35, 58 (FASEB meeting abstract), 59, 61(remove e13)) that are incomplete. Some of these papers have id numbers instead of page numbers.  The paper id info should be included. Some references need to have the correct format, e.g. year in bold.

Figs 1&2 and associated text on line 113 need proper citation. ref 32 or ref 24 only says “Experiments were performed as described in [32]” (or in [24]), but the data are not found in these papers.

Please check through the entire text and use “G-protein” or “G protein” consistently.

Author Response

Development of MOR agonists that are efficacious but have less adverse effects are challenging. This manuscript provides a timely summary of the current research data, especially on a few notable new molecular entities, such as oliceridine, PZM21 and SR-17018. The text is clearly written, and literature coverage up to date. The main conclusion and cautious recommendation are well balance. Overall, the manuscript has good educational value.

We would like to thank Reviewer #2 for his/her comments, suggestions and nice words (text clearly written, main conclusion and cautious recommendation are well balance, good educational value) regarding our manuscript.

Since oliceridine is one of the key compounds under discussion, I would urge the authors to wait for the PDUFA announcement from FDA which is due on August 7th and make appropriate revision based on the conclusion and disclosed new data.

The FDA approved oliceridine with the name Olinvyk™ on 7th August 2020 for short term intravenous use in “hospitals and other controlled settings”. This important information has been included in the revised version of the manuscript (pag 7, line 247).

Reference 69 is a comprehensive introduction on oliceridine, hence its citation should be moved to earlier place where oliceridine is mentioned rather than the current place in Conclusions.

As requested by this Reviewer the ref 69 of the original version has been moved where oliceridine is first mentioned in the revised version of the manuscript (pag 6, line 178).

There are a few references (e.g. 2, 8, 12, 30, 31, 32, 35, 58 (FASEB meeting abstract), 59, 61(remove e13)) that are incomplete. Some of these papers have id numbers instead of page numbers.  The paper id info should be included. Some references need to have the correct format, e.g. year in bold.

In the revised version of the manuscript the references have been carefully checked.

Figs 1&2 and associated text on line 113 need proper citation. ref 32 or ref 24 only says “Experiments were performed as described in [32]” (or in [24]), but the data are not found in these papers.

Data summarized in figs 1 and 2 are original findings from our laboratory. This has been made clearer in the revised version of the manuscript.

Please check through the entire text and use “G-protein” or “G protein” consistently.

“G protein” has been consistently used in the revised version of the manuscript.

Reviewer 3 Report

This reviewer thinks that this short review article is interesting and thought-provoking.

The aim of this brief review article was to analyze the available literature to determine whether mu-opioid receptor agonists biased toward G protein-coupled receptor signaling pathways are potential innovative analgesics, which would have therapeutic effects with decreased adverse effect profiles.

"Significant plagiarism" from lines 44-69 was detected via grammarly.com. This reviewer asks that the authors please revise this section. No other plagiarism was detected. 

There were several grammatical errors but overall, this reviewer thinks that the article has merit and is publishable after revision.

Please see this reviewer’s attached edits in a Word document to facilitate cutting and pasting.

Author Response

Extensive editing of English language and style required

This reviewer thinks that this short review article is interesting and thought-provoking.

The aim of this brief review article was to analyze the available literature to determine whether mu-opioid receptor agonists biased toward G protein-coupled receptor signaling pathways are potential innovative analgesics, which would have therapeutic effects with decreased adverse effect profiles.

We would like to thank Reviewer #3 for his/her comments, suggestions and nice words (interesting and thought-provoking article) regarding our manuscript. English language and style have been improved in the revised version of the manuscript.

"Significant plagiarism" from lines 44-69 was detected via grammarly.com. This reviewer asks that the authors please revise this section. No other plagiarism was detected.

We were unable to find any plagiarism in the mentioned text, however this has been modified in the revised version of the manuscript.

There were several grammatical errors but overall, this reviewer thinks that the article has merit and is publishable after revision.

Please see this reviewer’s attached edits in a Word document to facilitate cutting and pasting.

Thanks a lot for revision work made by this Reviewer; most if not all the edits of the word document have been included in the revised version of the manuscript

Reviewer 4 Report

I congratulate the authors with an excellent review on a highly relevant topic. I was preparing a similar review but feel that this review fills the need and no additional review is needed. I give the authors some minor items that they may consider.

Regarding the measurement of respiratory depression. Many animal studies claim that they measured respiratory depression but such measurements are difficult. Many results presented in the literature are therefore at best suspicious. For example, saturation is a measure of gas exchange and not ventilation and fully dependent on flow through the lungs. So I have a problem with the Manglik 2016 paper (Nature), see their Fig 4g and focus on vehicle. How come vehicle is a respiratory depressant? With reduction in respiratory rate by 200 breaths/min.This clearly is a methodical issue that gives me some problems with undetsnding the remainder of results (on respiration). 

Oliceridine (please use either oliceridie or TRV130 but do not use both, eg page 8, sec paragraph). We recently reanalyzed the human oliceridine data using satiny functions, see Dahan et al. Anesthesiology 2020, pub ahead of print; The paper addresses the mechanism of oliceridine in producing less respiratory depression than morphine (bias or partial agonism). For clinical practice I do not think this is an important issue.

I would also like to point the authors towards Olofsen et al. Anesthesiology 2019 on R-dihydro-etorphine. This is a mu/kappa/delta agonist and produces partial agonism on respiratory effect. However, this drug is biased towards the beta arresting pathway and therefore contradicts many of earlier findings on bias agonism.

Just some thoughts while reading your excellent paper. Thank you.

Author Response

I congratulate the authors with an excellent review on a highly relevant topic. I was preparing a similar review but feel that this review fills the need and no additional review is needed. I give the authors some minor items that they may consider.

We would like to thank Reviewer #4 for his/her comments, suggestions and nice words (excellent review on a highly relevant topic).

Regarding the measurement of respiratory depression. Many animal studies claim that they measured respiratory depression but such measurements are difficult. Many results presented in the literature are therefore at best suspicious. For example, saturation is a measure of gas exchange and not ventilation and fully dependent on flow through the lungs. So I have a problem with the Manglik 2016 paper (Nature), see their Fig 4g and focus on vehicle. How come vehicle is a respiratory depressant? With reduction in respiratory rate by 200 breaths/min. This clearly is a methodical issue that gives me some problems with undetsnding the remainder of results (on respiration).

We do agree with the Reviewer on this point which has been mentioned in the revised version of the manuscript (pag 7, line 272). 

Oliceridine (please use either oliceridie or TRV130 but do not use both, eg page 8, sec paragraph). We recently reanalyzed the human oliceridine data using satiny functions, see Dahan et al. Anesthesiology 2020, pub ahead of print; The paper addresses the mechanism of oliceridine in producing less respiratory depression than morphine (bias or partial agonism). For clinical practice I do not think this is an important issue.

The name oliceridine has been consistently used in the revised version of the manuscript and the ahead of print article by Dahan et al has been quoted (pag 7, line 356, ref 49).

I would also like to point the authors towards Olofsen et al. Anesthesiology 2019 on R-dihydro-etorphine. This is a mu/kappa/delta agonist and produces partial agonism on respiratory effect. However, this drug is biased towards the beta arresting pathway and therefore contradicts many of earlier findings on bias agonism.

Just some thoughts while reading your excellent paper. Thank you.

This information is possibly interesting however the involvement of different receptor types in the action of the molecule makes the interpretation of the results difficult.